# A computational model of pig ventricular cardiomyocyte electrophysiology and calcium handling: Translation from pig to human electrophysiology

Namit Gaur[1,2], Xiao-Yan Qi[3], David Benoist[1,4], Olivier Bernus[1,4], Ruben Coronel[1,5], Stanley Nattel[1,3,6], Edward J. Vigmond[1,2]*

1 IHU Liryc, Electrophysiology and Heart Modeling Institute, Fondation Bordeaux Université, Pessac-Bordeaux, France, 2 Univ. Bordeaux, IMB, UMR 5251, Talence, France, 3 Montreal Heart Institute and Université de Montréal, Montreal, Canada, 4 Univ. Bordeaux, Inserm, CRCTB, U1045, Pessac, France, 5 Department of Experimental Cardiology, Academic Medical Center, Amsterdam, The Netherlands, 6 Institute of Pharmacology, West German Heart and Vascular Center, Faculty of Medicine, University Duisburg-Essen, Essen, Germany

* edward.vigmond@u-bordeaux.fr

**Data Availability Statement:** All relevant data are within the manuscript and its Supporting Information files.

## Abstract

The pig is commonly used as an experimental model of human heart disease, including for the study of mechanisms of arrhythmia. However, there exist differences between human and porcine cellular electrophysiology: The pig action potential (AP) has a deeper phase-1 notch, a longer duration at 50% repolarization, and higher plateau potentials than human. Ionic differences underlying the AP include larger rapid delayed-rectifier and smaller inward-rectifier K⁺-currents ($I_{Kr}$ and $I_{K1}$ respectively) in humans. AP steady-state rate-dependence and restitution is steeper in pigs. Porcine $Ca^{2+}$ transients can have two components, unlike human. Although a reliable computational model for human ventricular myocytes exists, one for pigs is lacking. This hampers translation from results obtained in pigs to human myocardium. Here, we developed a computational model of the pig ventricular cardiomyocyte AP using experimental datasets of the relevant ionic currents, $Ca^{2+}$-handling, AP shape, AP duration restitution, and inducibility of triggered activity and alternans. To properly capture porcine $Ca^{2+}$ transients, we introduced a two-step process with a faster release in the t-tubular region, followed by a slower diffusion-induced release from a non t-tubular subcellular region. The pig model behavior was compared with that of a human ventricular cardiomyocyte (O'Hara-Rudy) model. The pig, but not the human model, developed early afterdepolarizations (EADs) under block of $I_{K1}$, while $I_{Kr}$ block led to EADs in the human but not in the pig model. At fast rates (pacing cycle length = 400 ms), the human cell model was more susceptible to spontaneous $Ca^{2+}$ release-mediated delayed afterdepolarizations (DADs) and triggered activity than pig. Fast pacing led to alternans in human but not pig. Developing species-specific models incorporating electrophysiology and $Ca^{2+}$-handling provides a tool to aid translating antiarrhythmic and arrhythmogenic assessment from the bench to the clinic.

**Funding:** NG,OB,ST and EV received financial support from the French Government as part of the "Investments of the Future" program managed by the National Research Agency (ANR), Grant reference ANR-10-IAHU-04 (www.anr.fr) EV,OB, and RC received funding from the Fondation Leducq, Research Grant number 16 CVD 02 (https://www.fondationleducq.org). The funders had no role in study design, data collection and analysis, decision to publish, or preparation of the manuscript.

**Competing interests:** The authors have declared that no competing interests exist.

## Author summary

The pig is an animal commonly used experimentally to study diseases of the heart, as well as investigate therapies to treat them, such as drugs. However, although similar, pigs differ from humans in certain aspects which may mean experimental results do not always directly translate between species. We propose a mathematical model of porcine electrophysiology which can serve as a tool to understand differences between the species and translate responses. Using new measurements along with values from literature, we built a computer model of porcine cardiac myocyte which replicated voltage and calcium behaviour over a range of pacing frequencies. The pig cell had a two-stage calcium release, unlike humans with a single stage. We predict that pigs and humans differ in the type of potassium current block that makes them most susceptible to cardiac arrhythmia. The model we developed can elucidate important differences between human and pig arrhythmia response.

## Introduction

The pig is commonly used in experimental models to study mechanisms of life-threatening arrhythmias that lead to sudden cardiac death (SCD) in humans [1–4], and to analyze the role of modulating factors (pharmacological [5], genetic [6], dietary [7–9], neural [10]), because its cardiac action potential (AP) duration (APD) and heart size are similar to those of humans.

Computational models of ventricular cardiomyocyte APs have been very successful in explaining arrhythmia mechanisms, altered function and drug therapy mechanisms during cardiac pathologies such as myocardial infarction (MI) [11], heart failure [12], gene mutations affecting ion channel subunits [13], drug therapy [14] and ischemia/hypoxia [15]. These models integrate multiple experimental data-sets of ion channel, ion pump and exchanger function, as well as intracellular ionic sub-compartment handling in ventricular myocytes. The underlying data are often derived from human but also from other species. However, at present, a single-cell ionic model of pig ventricular APs does not exist, despite the widespread use of pigs in preclinical cardiac research. Thus, a potentially valuable tool in the integration of pig electrophysiology and its translation to human arrhythmogenesis is lacking. Ion-channel function underlying AP morphology, $Ca^{2+}$-handling and other intracellular processes are species-specific. These species-specific physiological differences can affect the conditions leading to single-cell arrhythmogenic activity (*e.g.* early afterdepolarizations (EADs), delayed afterdepolarizations (DADs) and triggered activity (TA)) as well as AP rate-dependent phenomena like APD restitution and AP alternans. As a consequence, without computational models to understand the underlying differences between the cardiomyocytes of these two species translating results is very difficult.

The goal of the present study was to develop a mathematical model of the pig ventricular AP based on experimental data. Experimental AP characteristics like morphology, APD, AP amplitude (APA), resting membrane potential (RMP), $Ca^{2+}$-handling properties such as the rate dependence of intracellular $Ca^{2+}$-transients ($[Ca^{2+}]_i$) and directly-measured ion-current properties were used to calibrate the model.

Subsequently, the electrophysiology and $Ca^2$-handling of the pig single-cell model were compared to those of a widely used human ventricular AP model (O'Hara-Rudy (ORd)) [16]. In addition, steady-state rate-dependence of porcine APD, S1S2 APD restitution, and

alternans were compared to human, along with the inducibility of arrhythmogenic EADs, DADs and TA.

## Results

### Development of the pig ventricular AP model

The ionic current formulations in the pig ventricular AP model were developed based on available [8] and new experimental data (Fig 1). These new data include steady-state voltage activation/inactivation dependence of $I_{Na}$ (Fig 1A). Experimentally-measured voltage-dependence of (in)activation of $I_{Na}$ were fit with the required $I_{Na}$ steady-state (in)activation gating parameters (for m, h gates of $I_{Na}$) to yield half-maximal voltage ($V_{1/2}$) and slope factors (Fig 1A). The I-V curve, steady-state inactivation and recovery from voltage-dependent inactivation of $I_{CaL}$ were used to calibrate the model parameters of $I_{CaL}$: $V_{1/2}$ of activation, inactivation and time constant of voltage-dependent inactivation (Fig 1B–1D). E-4031 sensitive current during depolarizing pulses from -50 mV to +40 mV indicated $I_{Kr}$ (Fig 1E) and chromanol 293B-sensitive current $I_{Ks}$ (Fig 1F). The model-generated I-V curves of $I_{Kr}$ and $I_{Ks}$ matched well with experimental results (Fig 1E and 1F). $I_{Ks}$ was measured with 4s pulses to various voltages from -60 mV. The activation time constants were matched to experimentally-determined values (Fig 1G). $I_{K1}$ was fit to steady-state $Ba^{2+}$-sensitive current during hyperpolarizing pulses from -40 mV to various potentials for 300 ms (Fig 1H).

$I_{CaL}$, $I_{K1}$ and $I_{to2}$ conductances were adjusted to match experimental AP morphology recorded at 1 Hz [8] (Fig 2A). With these settings, at 1 Hz the model $APD_{90}$ was 293 ms with RMP -86.3 mV (Fig 2B). $I_{Na}$ conductance was determined by matching the maximal rate of experimental AP upstroke (inset Fig 2B). Our measured pig intracellular $[Ca^{2+}]_i$ transients displayed a two step increase (Fig 2C). We assumed a faster first $Ca^{2+}$ rise due to release from the sarcoplasmic reticulum (SR) in the T-tubular region, and a delayed $Ca^{2+}$-diffusion-mediated SR $Ca^{2+}$ release from non-T-tubular regions [17]. This feature was incorporated into the $Ca^{2+}$-handling components of the model. Model $[Ca^{2+}]_i$ at 0.5–2 Hz matched the morphology of experimental $[Ca^{2+}]_i$ (Fig 2C and 2D)[8], and had a positive frequency relationship. The model was also fit against $APD_{20}$, $APD_{50}$, $APD_{90}$ at 1 Hz [8] (Fig 2E). At 1 Hz, comparison of model and experimental RMP and APA is shown (Fig 2F). The APD restitution curve was also adjusted to experimental data [8] (Fig 2G). We next compared the behavior of the pig AP model to the human AP model.

### AP-driven currents

In Fig 3, a model pig ventricular AP is compared to a model human ventricular AP at a CL of 1000 ms, together with the associated ionic transmembrane currents. The porcine AP differs from the human AP, with a slightly deeper notch, a higher AP plateau and faster terminal repolarization than the human AP. The deeper notch is caused by larger transient outward current ($I_{to}$) in pig compared to the endocardial human model (Fig 3H). $I_{to}$ in pig is due to $Ca^{2+}$-activated $Cl^-$ current ($I_{to,2}$) [18] while in human, it is due to $K^+$ current($I_{to,1}$). $I_{Na}$ is similar between these species but $I_{CaL}$ has a higher peak and a smaller sustained plateau current (Fig 3C and 3E). Similarly, peak porcine $I_{NaCa}$ is greater than human (Fig 3G). The repolarizing current $I_{Kr}$ is larger in human than pig (Fig 3B), but $I_{Ks}$ is larger in pig than human (Fig 3D). A greater $I_{K1}$ (Fig 3F) causes faster final repolarization in pig. $I_{Ks}$ is smaller compared to $I_{Kr}$ and $I_{to}$ in both species. Therefore, the differences in AP in pig and human correspond to different magnitudes of depolarizing ($I_{Na}$, $I_{CaL}$, $I_{NaCa}$) and repolarizing ($I_{Kr}$, $I_{Ks}$, $I_{to}$, $I_{K1}$) currents.

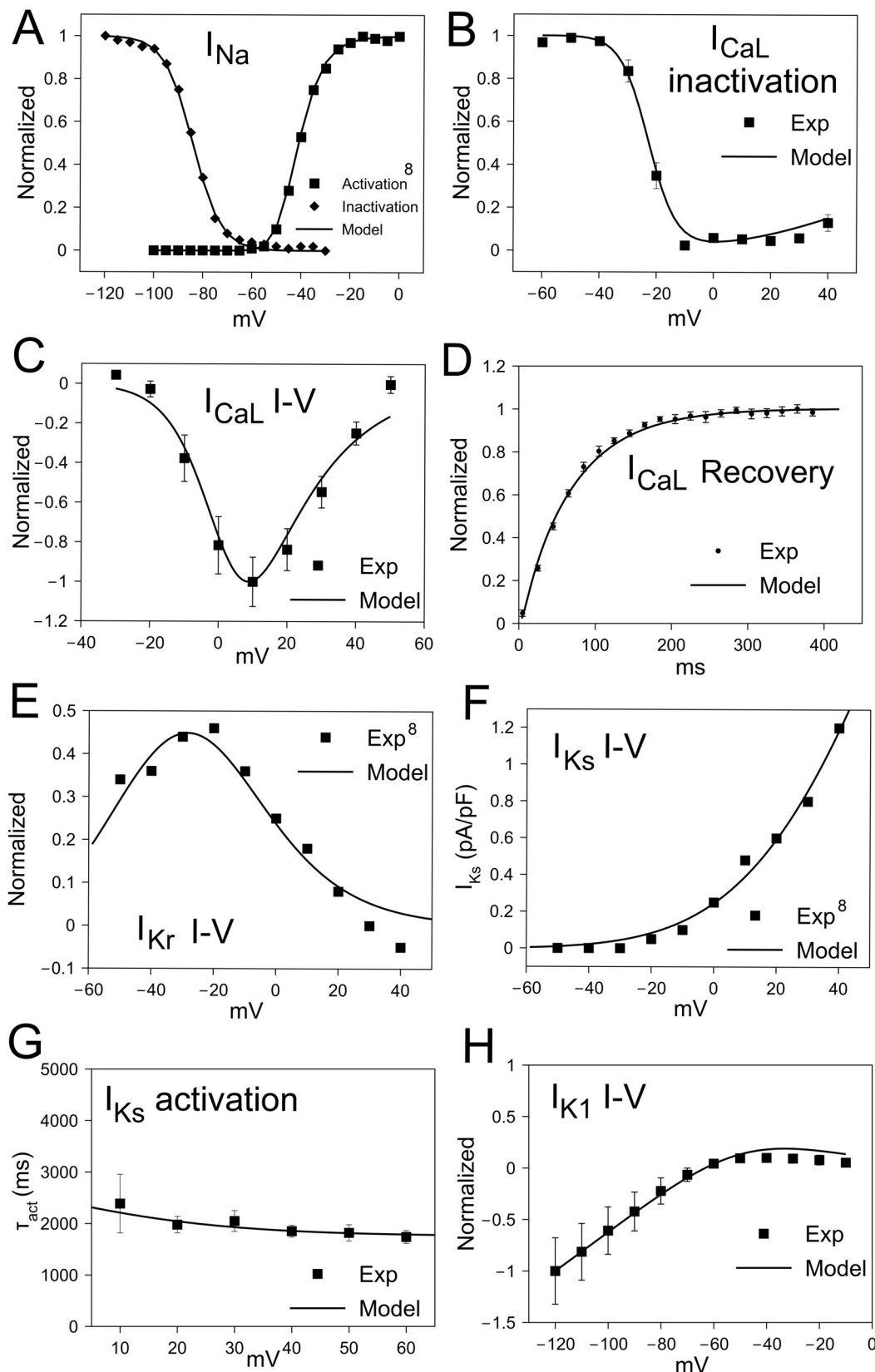

**Fig 1. Electrophysiological validation of pig ventricular myocyte action potential (AP) model.** (A) Steady-state activation/inactivation curves of Fast Na$^+$ current ($I_{Na}$). (B) Steady-state inactivation of L-type Ca$^{2+}$ current ($I_{CaL}$). (C) Current-voltage (I-V) relationship of $I_{CaL}$. (D) Recovery of $I_{CaL}$ after voltage-inactivation (n = 5 cells). (E) I-V curve of rapid delayed rectifier K$^+$ current ($I_{Kr}$). (F) I-V curve of slow delayed rectifier K$^+$ current ($I_{Ks}$). (G) Time constant of activation ($\tau_{act}$) of $I_{Ks}$ (n = 6 cells). (H) I-V curve of inward rectifier K$^+$ current ($I_{K1}$) (n = 5 cells). All model equations are included in the Data Supplement.

## Ca$^{2+}$ handling and ionic concentrations during pacing

Fig 4 shows Ca$^{2+}$ handling and ionic concentrations in pig and human ventricular myocyte models during 1 Hz pacing. [Ca$^{2+}$]$_i$ is smaller in pig (Fig 4A) and shows a two-step increase due to Ca$^{2+}$ release from t-tubular and non t-tubular subcellular regions [17]. Our pig model had network and junctional SR [Ca$^{2+}$] ([Ca$^{2+}$]$_{NSR}$ and [Ca$^{2+}$]$_{JSR}$) that were lower than in the human model (Fig 4C and 4E), but total (combination of t-tubular and non t-tubular) peak SR Ca$^{2+}$ release ($J_{rel}$) was larger than in the human model (Fig 4B). Peak subspace [Ca$^{2+}$] ([Ca$^{2+}$]$_{SS}$) in the t-tubular region was larger in pig (Fig 4G) and myoplasmic [Na$^+$] ([Na$^+$]$_i$) and [K$^+$] ([K$^+$]$_i$) is smaller (Fig 4F and 4H) than in human. Peak SR Ca$^{2+}$ uptake ($J_{up}$) was of similar magnitude but showed a two phase component (Fig 4D).

## Rate-dependence and restitution of APD

The steady-state APs at various cycle lengths (CLs) ranging from 400–5000 ms are shown in Fig 5A and 5B. The maximum slope of the steady-state APD rate-dependence (adaptation) is steeper in pig (Fig 5C) than in human, indicating a greater rate-adaptation response. The S1S2 APD restitution (S1 = 1000 ms) is also steeper in pig compared to human (Fig 5D).

## Afterdepolarizations and triggered activity

Blockade of several repolarizing currents (especially $I_{Kr}$) is an off-target effect of many pharmaceutical compounds [19], but reduced repolarizing currents can also result from gene mutations. Decreased repolarizing current can lead to APD prolongation and increased risk of QT interval prolongation and Torsade de Pointes (TdP) arrhythmia [19]. In order to assess differential AP prolongation/shortening and repolarization abnormalities in human and pig, we simulated block of each of the repolarizing currents: $I_{Kr}$, $I_{K1}$, $I_{Ks}$ and $I_{to}$ by decreasing their conductance values. Up to 90% block of $I_{Ks}$ and $I_{to}$ did not cause repolarization abnormalities in the form of EADs in either human or pig although the % APD prolongation was larger in pig ($I_{Ks}$ block: 14.3%/4.5% pig/human; $I_{to}$ block: 5.6%/1% pig/human). In contrast, block of $I_{Kr}$(conductance reduced by 90%) caused EADs in human but not in pig (both CL = 2000 ms; Fig 6A and 6B). Block of $I_{K1}$ led to EADs in pig but not in human (20% of normal conductance, Fig 6C and 6D). These differential results of block of repolarizing currents indicate that the pig is more susceptible to EAD formation due to $I_{K1}$ block while human is to $I_{Kr}$ block. Thus, proarrhythmic intervention in pigs are not necessarily proarrhythmic in humans, and drugs without proarrhythmic effects in pigs are not necessarily safe in humans.

The pig cell model predicted lower [Ca$^{2+}$]$_{NSR}$, [Ca$^{2+}$]$_{JSR}$ and [Ca$^{2+}$]$_i$ compared to human (Fig 4A, 4C and 4E). To produce DADs after cessation of pacing at 1 Hz (Fig 7A), the threshold of spontaneous SR Ca$^{2+}$ release was set to 2 mM in human and 1.25 mM in pig. The last paced steady-state beat at CL = 1000 ms is indicated by arrow. The appearance of subthreshold DADs was similar (5 occurred in human and 7 in pig during 10s post-pacing). The mechanism of DAD formation in both human and pig is activation of $I_{NaCa}$ resulting in a net inward current (Fig 7D) secondary to spontaneous SR Ca$^{2+}$ release (Fig 7C) similar to observations in experiments [20, 21].

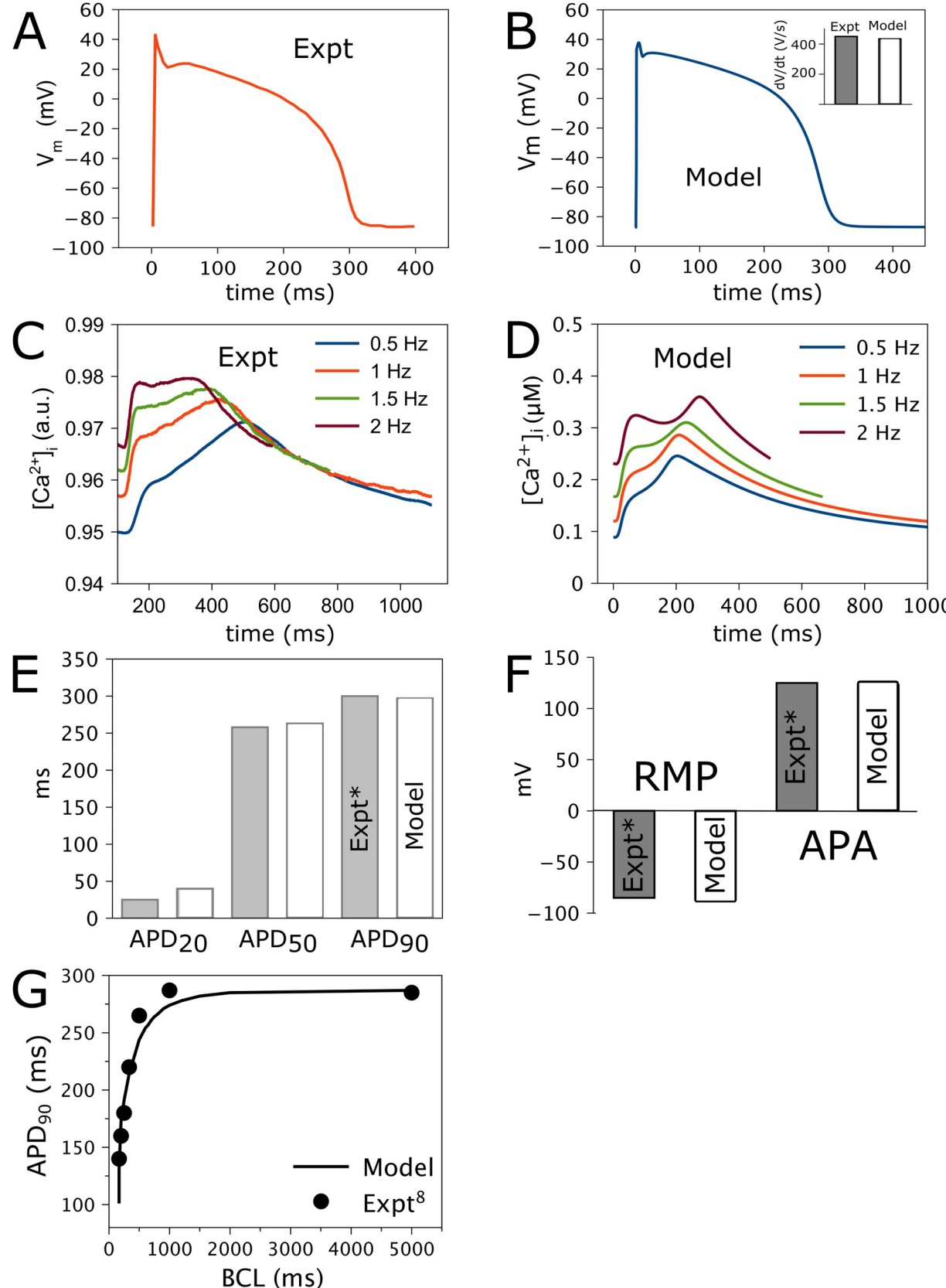

**Fig 2. Ca$^{2+}$ handling and action potential (AP) validation of pig ventricular myocyte AP model.** (A) Experimental AP morphology at 1 Hz pacing. (B) Model AP at 1 Hz. (C) Experimental Fluorescence values of intracellular Ca$^{2+}$ transients ([Ca$^{2+}$]$_i$) at four pacing rates: 0.5, 1, 1.5 and 2 Hz. (D) Average of [Ca$^{2+}$]$_i$ in the t-tubular and non t-tubular subcellular regions at these rates. (E) Experimental and model AP duration at 20, 50 and 90% repolarization (APD$_{20}$, APD$_{50}$, APD$_{90}$). (F) Experimental and model resting membrane potential (RMP) and AP amplitude (APA) at 1 Hz pacing.

For the same simulation shown, but when the I$_{Na}$ activation curve was shifted leftwards by 10 mV, triggered activity (TA, indicated by $^*$) occurred in the human cell (Fig 7E). The last paced beat is indicated by the arrow. TA in pig cells required leftward I$_{Na}$ activation by 15 mV as well as destabilizing the resting membrane potential by reducing I$_{K1}$ conductance by 50%. The frequency of TA was similar (7 in 4s for human; 5 for pig). TA was activated in both human and pig when the threshold membrane potential of I$_{Na}$ activation was reached after I$_{NaCa}$ was activated (Fig 7H) by spontaneous SR Ca$^{2+}$ release (Fig 7G). I$_{to2}$ has been shown to play a role in DADs [22], but porcine I$_{to2}$ was not a contributor to DAD formation in these simulations. The need for both a greater leftward I$_{Na}$ activation shift and a reduced I$_{K1}$ conductance to produce TA in pig cardiomyocytes shows that they are less susceptible to TA induced by DADs than human myocytes.

## AP and Ca2+ transient alternans

Cytosolic calcium concentration was measured in isolated porcine biological myocytes paced at increasing frequencies from 1 to approximately 5 Hz in several stages (Fig 8A). Note that these myocytes were not used for previous model fitting, nor was data obtained from pacing faster than 2 Hz used for fitting. No alternans occurred at any pacing frequency, neither transiently nor at steady state. The Ca$^{2+}$-transient morphology changed with rate, with the diastolic and peak systolic levels monotonically increasing with rate. For slow rates, the rapid Ca$^{2+}$ upstroke was followed by a more gradual increase over 100 ms. As pacing frequency increased, the slower rise leveled off (1.6 and 2.2 Hz) and then became a slower decrease (>2.2 Hz). At 5 Hz, the transient was triangular.

A similar protocol was carried out *in silico* for human and pig models. Qualitatively, the pig results matched biological experiment (Fig 8B). Systolic and diastolic levels increased with pacing rate. Importantly, alternans was not seen, neither in [Ca$^{2+}$]$_i$ nor in V$_m$. The computational Ca$^{2+}$-transient experienced the same frequency-dependent changes in morphology, albeit at different frequencies from the experimentally recorded transients. There are two noticeable differences: 1) at 400 ms cycle length, systolic [Ca$^{2+}$]$_i$ dropped from the 500 ms cycle length. This is related to the merging of the two phases of the Ca-transient at higher heart rates. 2) Systolic and diastolic [Ca$^{2+}$]$_i$ rose dramatically from 300 to 200 ms cycle length, although a large rise in diastolic [Ca$^{2+}$]$_i$ was also seen experimentally. Although these likely represent limitations of the model at very high heart rates, it does qualitatively correctly replicate the increase in diastolic Ca$^{2+}$-concentration, and the Ca$^{2+}$-transient duration is well reproduced. Increasing the proportion of Ca$^{2+}$ release occurring in the t-tubules above 50% in the model created alternans and other periodicity in the morphology of Ca$^{2+}$ transients.

In the human model (Fig 8C), the response was different. The Ca$^{2+}$ transients retained the same morphology over all frequencies until 200-ms cycle-length pacing (5 Hz), at which point alternans appeared. There was a rise in diastolic [Ca$^{2+}$]$_i$, although it appeared to be smaller than in the pig. Very modest voltage alternans, on the order of a few mV, accompanied the [Ca$^{2+}$]$_i$ alternans.

Biological experiments were also performed with a sudden change in pacing frequency in an effort to evoke alternans. Cells were paced at 0.5 Hz until steady state and the rate abruptly

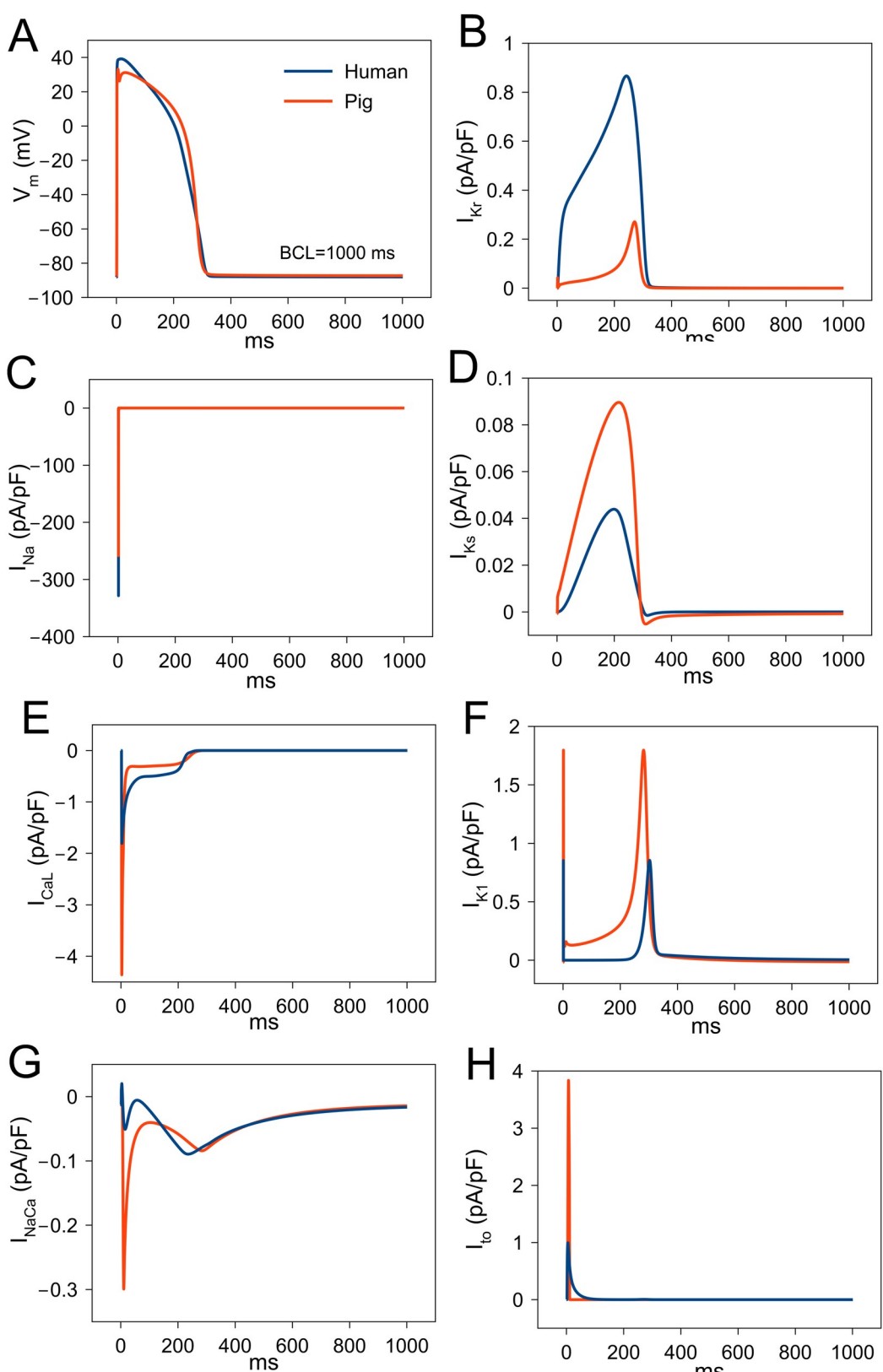

**Fig 3. Comparison of simulated cardiac electrophysiological currents between human and pig ventricular myocyte during paced action potential (AP) at a basic cycle length (BCL) of 1000 ms.** (A) AP. (B) Rapid delayed rectifier K$^+$

current ($I_{Kr}$). (C) Fast Na$^+$ current ($I_{Na}$). (D) Slow delayed rectifier K$^+$ current ($I_{Ks}$). (E) L-type Ca$^{2+}$ current ($I_{CaL}$). (F) Inward rectifier K$^+$ current ($I_{K1}$). (G) Total (combination of myoplasmic and subspace) Na$^+$-Ca$^{2+}$ exchanger current ($I_{NaCa}$). (H) Transient outward current ($I_{to}$). $I_{to}$ is due to Ca$^{2+}$-activated Cl$^-$ current ($I_{to2}$) in pig and due to K$^+$ current ($I_{to1}$) in human.

increased to 2 Hz. No alternans was seen in [Ca$^{2+}$]$_i$ in the experiment. Replication of the experiment in the pig and human computer models also failed to show alternans. Abruptly changing to a higher frequency did not result in loss of 1:1 capture in either model.

## Discussion

We developed a computational model of pig ventricular cardiomyocyte AP based on available and new experimental characterizations of cellular electrophysiology and Ca$^{2+}$ handling. Our study is the first to describe a computational model of pig ventricular myocyte AP and paves the way for integration of pig ventricular cardiomyocyte AP model into whole-heart models for future studies to test ideas about multicellular arrhythmogenic mechanisms (like reentry). The model can be used to test potential drug therapies for treating cardiac pathological conditions such as heart failure and allows translation of pig experimental data to human arrhythmogenesis. We report that alternans is difficult to induce in isolated pig ventricular cells.

We have used a subendocardial human ventricular cardiomyocyte model for comparison with pig APs in this study. However, our conclusions about differences in arrhythmogenesis between human and pig should not change if we had used a subepicardial or midmyocardial human model, since the major difference in transmural gradients of ionic current is in $I_{to}$. Moreover, the notch in subepicardial human AP is not much greater than the subendocardial human AP notch [16], suggesting relatively small differences in $I_{to}$. Although extensive cardiac electrophysiological data exist at the tissue/whole-heart scale for experimental pig models used to study arrhythmogenesis, limited data exist at the cell-scale. We have used available experimental data at the cell scale. Where data were missing, we performed new measurements of $I_{CaL}$, $I_{K1}$ and $I_{Ks}$ properties and used them for model calibration.

Our results show that 1) pig is more susceptible to AP-prolongation to $I_{K1}$ block while human is more susceptible to $I_{Kr}$ block, 2) pig cells are less susceptible to TA induced by DADs than human cardiomyocytes, and 3) human APs are more susceptible to alternans than pig APs. Furthermore, the steady-state rate-dependence of AP, *i.e.*, steady state adaptation, and S1S2 restitution curves are steeper in pig. The maximum slope of APD restitution is linked to increased arrhythmias (like ventricular fibrillation [VF]) at the organ scale [23–25]. Indeed, pigs are reported to be particularly susceptible VF under stress [26]. In addition, as suggested by our model, pigs have been found to be highly sensitive to ventricular proarrhythmia due to $I_{K1}$ blockade [4] and insensitive to $I_{Kr}$ block [27].

### Differences between simulated and measured ionic currents

Some of the model ionic currents during pig AP differ from those measured in control healthy pig cells in a study investigating heart failure [28] although APD, AP morphology and restitution curves correspond well. In particular, model $I_{CaL}$ peak is larger and sustained current smaller and model $I_{NaCa}$ is inward throughout the AP. However, similar to these biological experiments, model $I_{Ks}$ is smaller than the other major repolarizing currents $I_{K1}$ and $I_{Kr}$. $I_{K1}$ is larger than $I_{Kr}$, but $I_{to2}$ duration is smaller. In another study, K$^+$ currents and APs varied substantially from different regions of a pig heart [29]. Our measured data are mainly taken from right ventricular mid myocardium from male pigs [8]. The observed differences are likely explained by measurements from different regions of the heart (apex/base of right/left

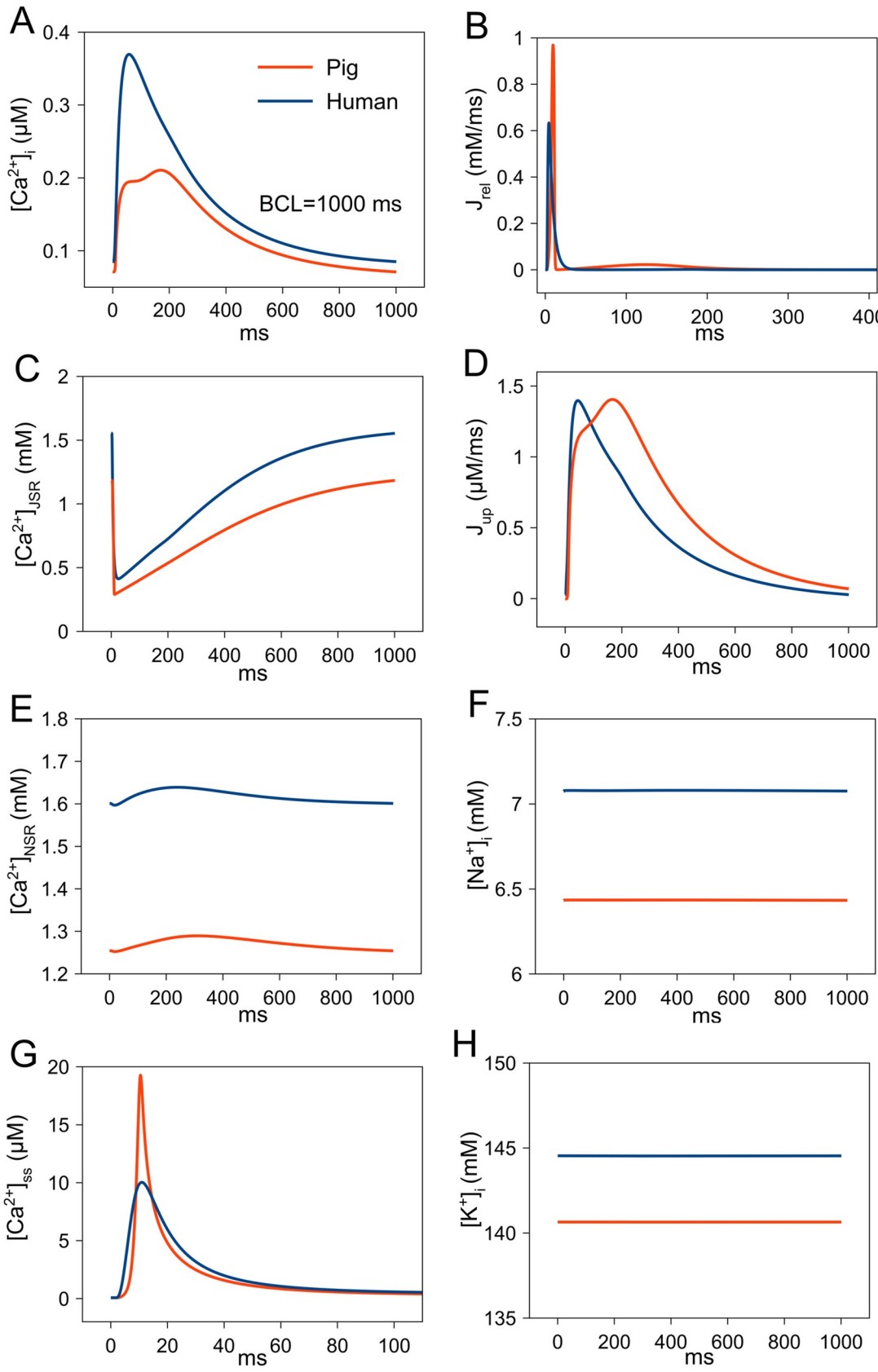

**Fig 4. Comparison of Ca²⁺ handling and ionic concentrations between human and pig ventricular myocyte during paced action potential (AP) at a basic cycle length (BCL) of 1000 ms.** (A) intracellular Ca²⁺ concentration (B) Sarcoplasmic Reticulum (SR) Ca²⁺ release flux ($J_{rel}$). (C) Junctional SR Ca²⁺ concentration ($[Ca^{2+}]_{JSR}$). (D) SR Ca²⁺ uptake flux ($J_{up}$). (E) Network SR Ca²⁺ concentration ($[Ca^{2+}]_{NSR}$). (F) Myoplasmic Na⁺ concentration ($[Na^+]_i$). (G) Subspace Ca²⁺ concentration ($[Ca^{2+}]_{ss}$). (H) Myoplasmic K⁺ concentration ($[K^+]_i$).

ventricle, subendocardium, subepicardium or midmyocardium), different experimental protocols or from the subspecies or sex of pig used [8,28,29].

Another reason for differences in our model is that we fit voltage and calcium data simultaneously over several pacing frequencies. A given model structure, which includes the physical geometry of the cell along with the mathematical forms of the equations used to represent the cellular components, is not guaranteed to reproduce the broad range of observations. Missing

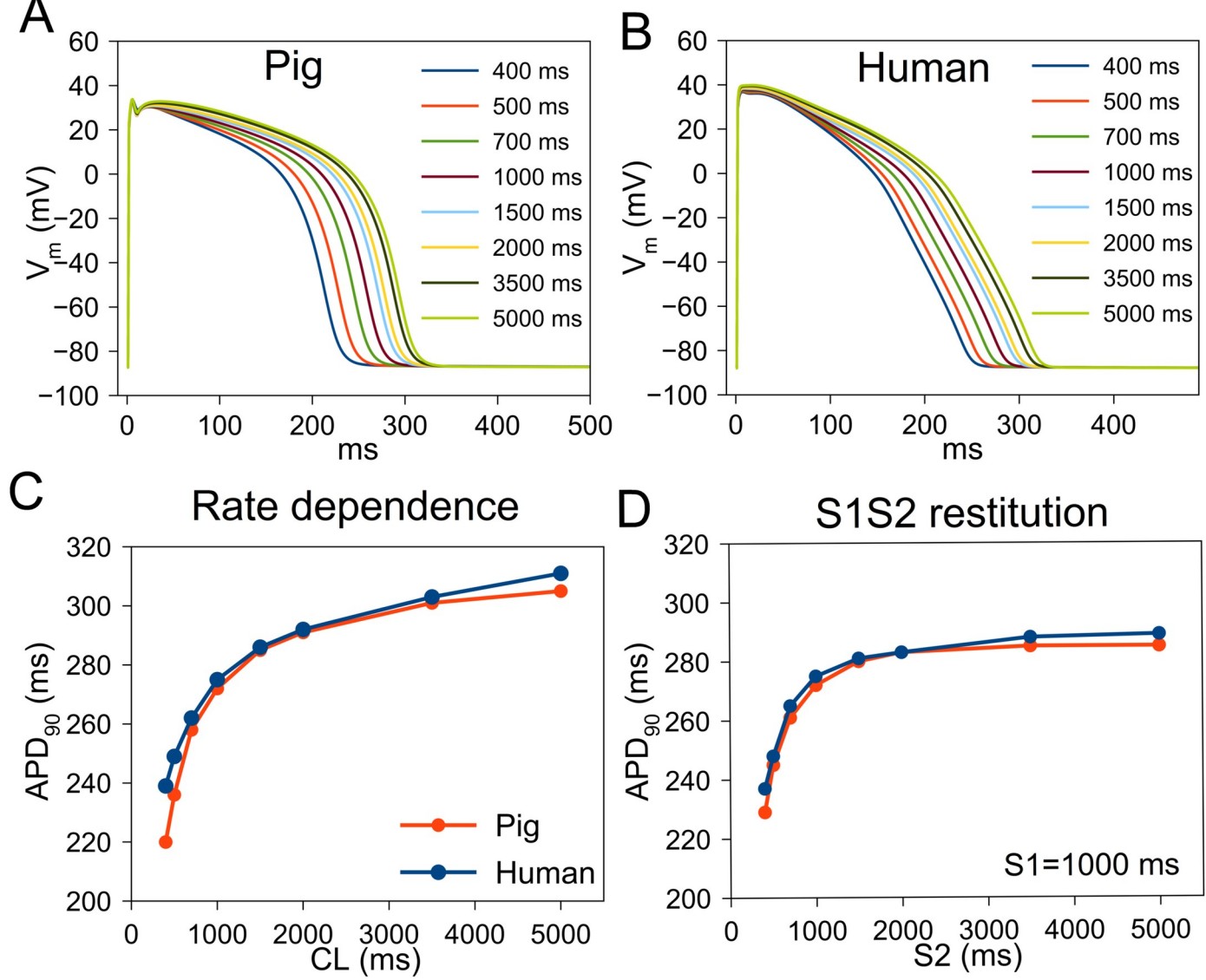

**Fig 5. Steady-state rate dependence (also called adaptation) and restitution of action potential (AP) duration (APD) in human and pig ventricular myocyte.** (A) Steady-state pig action potential (AP) at cycle length (CL) 400–5000 ms. (B) Human AP at CL 400–5000 ms. (C) Adaptation of APD at 90% repolarization ($APD_{90}$) in pig and human. (D) APD restitution in pig and human. The driving (S1) CL is 1000 ms followed by varying pacing intervals (S2).

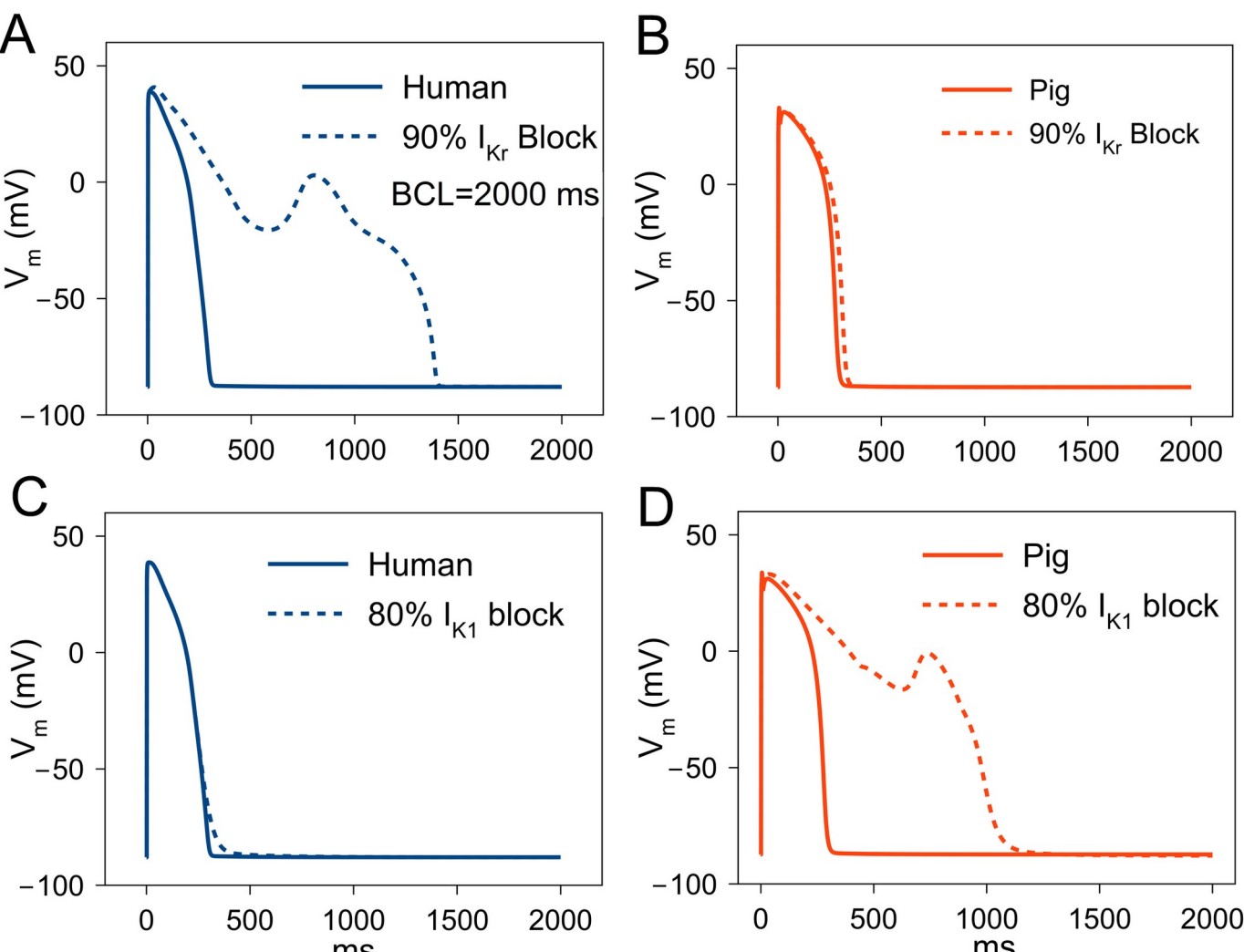

**Fig 6. Early afterdepolarization (EAD) formation in human and pig ventricular myocyte.** (A) 90% block of rapid delayed rectifier K$^+$ current (I$_{Kr}$) leads to EADs in human at a cycle length (CL) of 2000 ms. (B) Similar degree of block leads to small AP prolongation in pig. (C) 80% block of I$_{K1}$ leads to EADs in pig (D) but not in human.

biological processes or inappropriate mathematical formulations will limit the fidelity of the model. Improving the fitting of one aspect worsens another, and it is difficult to automate such multi-objective optimizations since the final choice is subjective according to the types of error the operator tolerates. For example, how much of a difference in AP restitution is allowed while ensuring that calcium transient waveforms are flat-topped at 500 ms pacing? Furthermore, given differences in individual cell behavior, our fit attempts to capture behavior within reasonable physiological limits.

## Afterdepolarizations

In human, I$_{Kr}$ block caused EADs, while in pig I$_{K1}$ block caused EADs. The balance of repolarizing currents shaping phase-3 and 4 of AP is in favor of I$_{K1}$ in pig compared to I$_{Kr}$ in human (Fig 3). I$_{Kr}$ is an important component of repolarization reserve [30] and larger I$_{Kr}$ in human can cause greater vulnerability to AP prolongation and Torsadegenic arrhythmias during

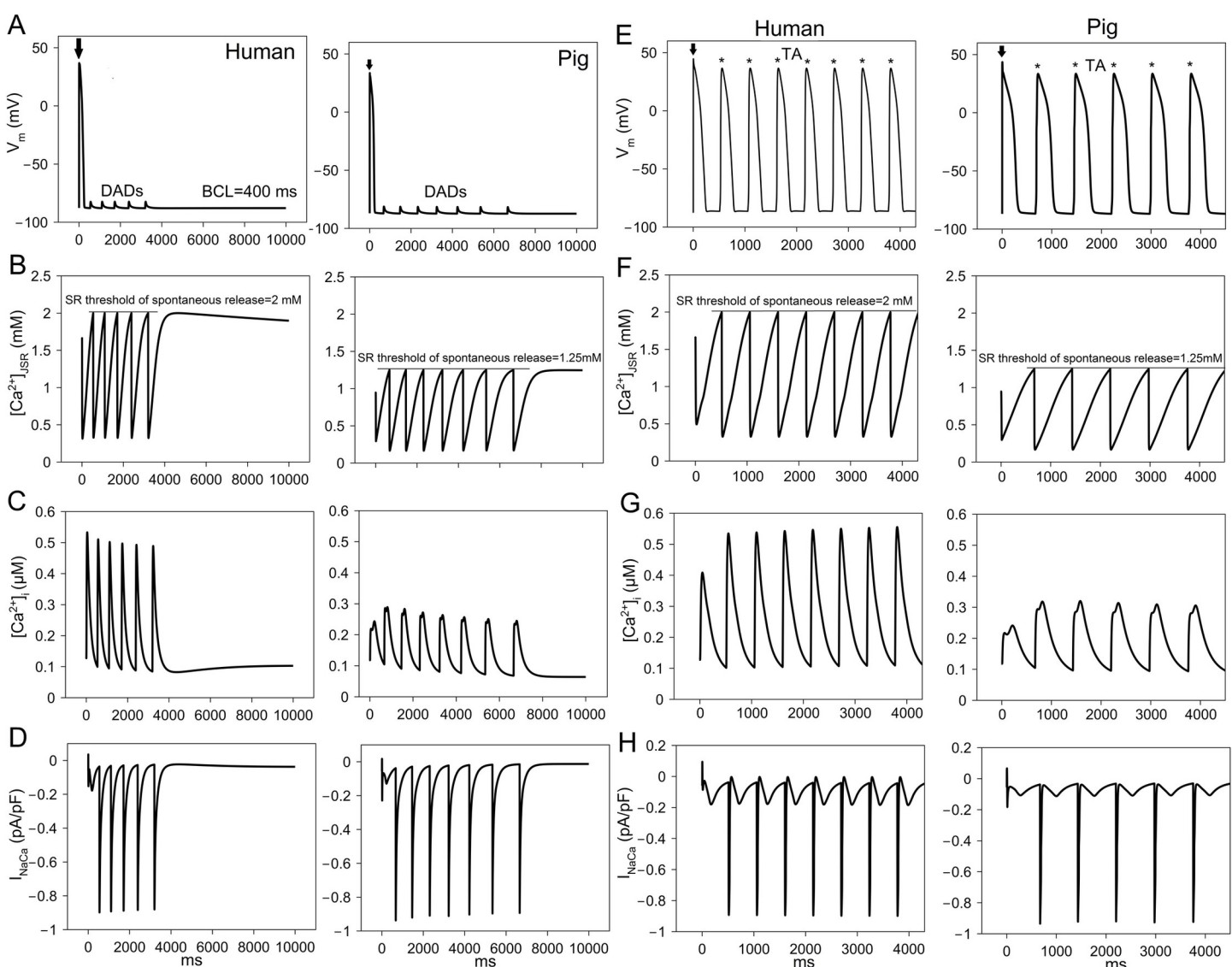

**Fig 7. Delayed afterdepolarizations (DAD) formation in human and pig ventricular myocyte.** (A) DADs in human and pig cells after cessation of pacing at cycle length (CL) = 400 ms. (B) Junctional sarcoplasmic reticulum (JSR) $Ca^{2+}$ concentration ($[Ca^{2+}]_{JSR}$). The threshold of spontaneous SR $Ca^{2+}$ release was 2 mM in human and 1.25 mM in pig. (C) Myoplasmic $Ca^{2+}$ concentration ($[Ca^{2+}]_i$). (D) $Na^+$-$Ca^{2+}$ exchanger current ($I_{NaCa}$). (E) Triggered activity (TA) occurred in human and pig cell after cessation of pacing at cycle length (CL) = 400 ms. The fast $Na^+$ current ($I_{Na}$) activation was shifted leftwards by 10 mV in human and 15 mV in pig. In addition, the inward rectifier $K^+$ current ($I_{K1}$) was reduced by 50% in the pig cell. The last paced beat is shown by an arrow. (F) Junctional Sarcoplasmic Reticulum (JSR) $Ca^{2+}$ concentration ($[Ca^{2+}]_{JSR}$). (G) Myoplasmic $Ca^{2+}$ concentration ($[Ca^{2+}]_i$). (H) $Na^+$-$Ca^{2+}$ exchanger current ($I_{NaCa}$). The spontaneous SR $Ca^{2+}$ release threshold was 2 mM in human and 1.25 mM in pig. Lines show the threshold level at which $Ca^{2+}$ release occurred.

repolarizing drug block. Larger $I_{K1}$ can be a significant contributor to repolarization reserve in pig and, indeed, its block also caused EADs in the pig cell model but not in the human cell model. $I_{K1}$ block has been reported to cause EADs in dog [31] and in a modeling study [32], as well as to be proarrhythmic in pig [4]. Conversely, the larger $I_{K1}$ in pig cardiomyocytes protected them from DADs, as shown by the requirement for 50% reduction in $I_{K1}$ to produce DADs in pig. $I_{Ks}$ plays a smaller role than $I_{K1}$ or $I_{Kr}$ in AP repolarization for both species, as has been shown earlier for human and canine models [33]. $I_{Ks}$ block did not prolong APD significantly, consistent with this observation. Similarly, $I_{to2}$ block did not prolong APD

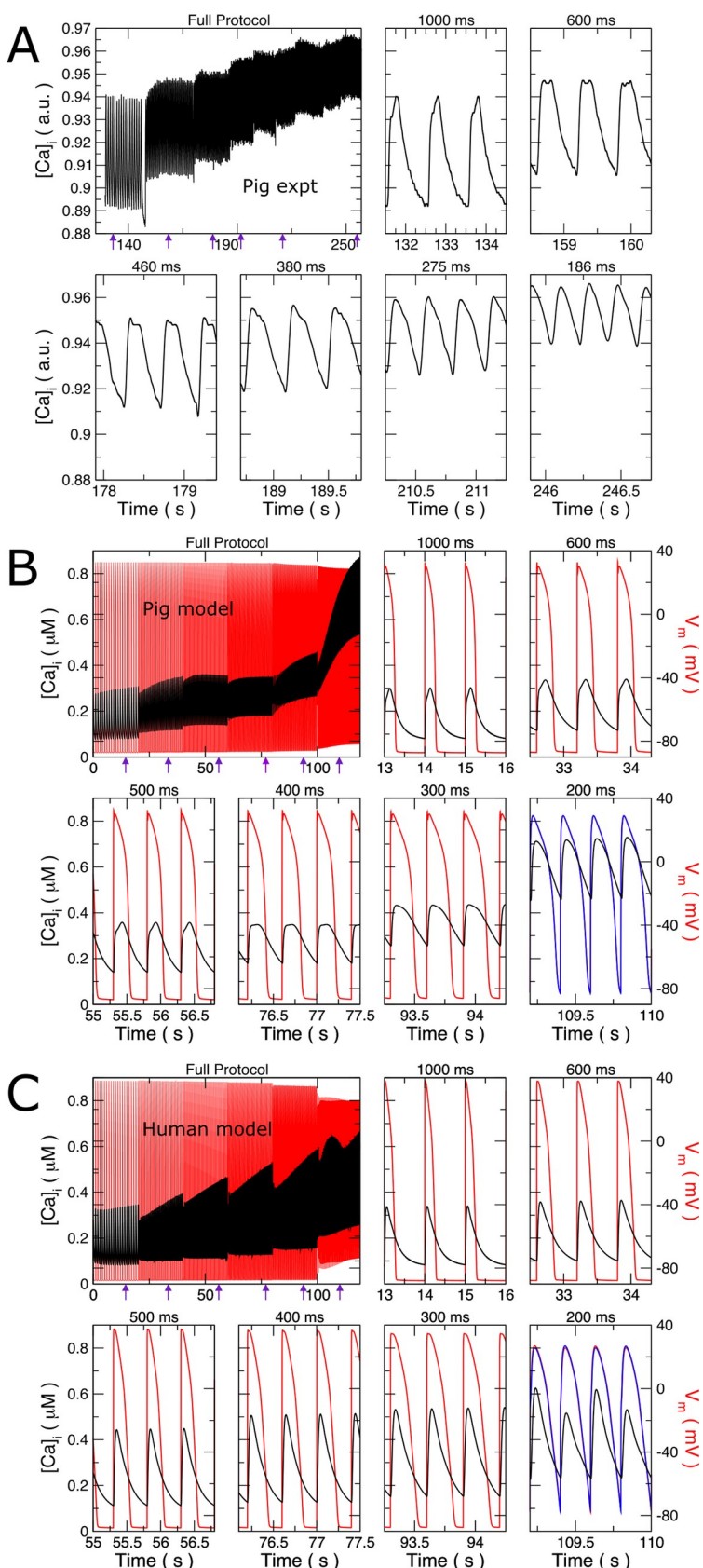

**Fig 8. Alternans produced in human but not in porcine model. Intracellular Ca$^{2+}$ transient [Ca]$_i$ and membrane voltage (V$_m$) response at various pacing frequencies.** (A) Relative [Ca]$_i$ of isolated right ventricular (RV) myocytes is shown versus time as the pacing cycle length progressively decreased in steps. The full pacing protocol is shown in the upper left graph. Excerpts are displayed with the exact pacing cycle length indicated above each. (B) Computer simulation of pig cell ionic model following the experimental protocol, but also showing transmembrane voltage (red). For 200 ms pacing, the transmembrane voltage trace shifted by one pacing cycle (blue) is also shown. Voltage alternans was not seen. (C) Computer simulation of human cell ionic model as in B. For 200 ms pacing, the transmembrane voltage trace shifted in time by one pacing cycle is also displayed, showing very small voltage amplitude alternans (1.2 mV). All x-axes display time. Purple arrows in the full protocol indicate the times of the excerpts.

significantly, but it abolished the AP notch. Abolishment of the AP notch can lead to lower driving force for I$_{CaL}$ and consequently smaller SR Ca$^{2+}$ release and lower contractility of the ventricles. I$_{K1}$ also sets the resting membrane potential of the cell. Thus, I$_{K1}$ block may promote arrhythmogenic DADs and TA besides EAD formation in pig.

After cessation of rapid pacing, DADs occurred in both human and pig cells. Human cells operated at higher Ca$^{2+}$ load at faster rates. For a comparable frequency of DADs (as shown in Fig 7), the SR threshold of spontaneous SR Ca$^{2+}$ release (= 1.25 mM) required was lower in pig. DADs occurred at a higher SR threshold in humans and for a comparable threshold to pig (= 1.25 mM), DAD frequency was higher in human. TA occurred in human and pig only if the I$_{Na}$-activation curve was shifted to the left. In pig, additional intervention of lower I$_{K1}$ (50% block) was required to generate TA; therefore, lower Ca$^{2+}$ load at fast rates, coupled with higher resting membrane stability (due to higher I$_{K1}$), makes pig cell less vulnerable to Ca$^{2+}$-dependent arrhythmogenesis linked to spontaneous Ca$^{2+}$ after transients. Although the absolute SOICR threshold was lower in the pig, which would suggest greater arrhythmogenic potential, porcine [Ca$^{2+}$]$_{NSR}$ and [Ca$^{2+}$]$_{JSR}$ were lower and the pig had stronger compensatory mechanisms to protect against DADs.

## Alternans

In the human model, APD alternans at faster rates (CL = 250 ms) coincided with instabilities in Ca$^{2+}$ cycling (Fig 8). Peak Ca$^{2+}$ transients alternated every other beat and Ca$^{2+}$ alternans drives electrical (AP) alternans [16]. In pig cardiomyocytes, alternans was not seen, either experimentally or in silico. This suggests that the less developed t-tubular network in pig may be protective against alternans development, as supported by the appearance of alternans in the pig model if the proportion of t-tubule release was increased. This study also suggests that alternans at the organ level in healthy porcine hearts might result from tissue and mechanical heterogeneity, and not from intrinsic cellular behavior.

In most mammalian ventricular cardiomyocytes, the density of t-tubules is uniform, resulting in synchronous calcium-induced calcium release from RyRs. In pigs, heterogeneities in t-tubule density correlates with variability in cell size [34]. Ca$^{2+}$ transients in pig ventricular cardiomyocytes have been reported to be single-peaked [8,28] or multi-peaked [9,17]. Our experimental data showed a distinct fast rising phase followed by a slower change in [Ca$^{2+}$]$_i$ transients (Fig 8A). We modeled the Ca$^{2+}$-handling system to fit these observations. Variability in cell size, cytoskeleton, Ca$^{2+}$-handling subsystem, or experimental protocols may account for these reported differences.

## Translating these results

Animal models are valuable for testing drugs and examining mechanisms. However, species differences can be significant and translating animal results to humans cannot be done directly. With our pig model, we will be better able to identify the mechanistic changes occurring experimentally, as well as quantify them. We can then implement these mechanistic

changes in a human model and assess their impact on the performance. It is entirely possible that an effect seen in one species is absent in another. An immediate conclusion from our study is that the effects of drugs affecting $I_{K1}$ (both antiarrhythmic and proarrhythmic) would be larger in pigs than in humans. For $I_{Kr}$, it would be the opposite. This study highlights differences in calcium handling between the species. Indeed, lack of DADs in porcine models due to the application of a drug does not necessarily indicate that the drug is safe in humans. Similarly, arrhythmogenic pathophysiological changes observed in porcine models do not necessarily translate to humans. Thus, our work highlights the caution needed in extrapolating between species and provides a tool with which to explore mechanisms of observed effects in pigs and their potential relationship to expected effects in humans.

## Limitations

The model uses whole-cell $Ca^{2+}$ handling formulations. Spatially-distributed models of $Ca^{2+}$-handling that can simulate stochastic $Ca^{2+}$-sparks and $Ca^{2+}$-waves were not considered because the computational costs of such simulations would have been prohibitive. This is especially relevant to consider when simulating porcine whole-heart electrical activity.

The mathematical formulations of ionic currents, pumps and exchangers were based on widely accepted biophysical mechanisms. There may be alternative formulations based on more complex biophysical models (such as Markov models of ion channels). We chose to select the simplest sufficiently detailed formulation that can fit the experimental data. Moreover, complex formulations generally involve larger numbers of parameters and consequently uncertainties in their quantitative determination. Though the model recreates experimental behavior qualitatively, there are quantitative differences. While APD restitution was closely reproduced by the model, the changes in Ca transient shape did not occur at the same frequency, and the exact $[Ca^{2+}]_i$ was not known. However, major behavioral characteristics were captured within physiological bounds. This model does represent the best approximation of pig ventricular myocyte cellular electrophysiology and calcium handling to date.

We used RV preparations from the pig for optical mapping of $Ca^{2+}$ to evaluate model predictions because of its favorable anatomy. For the ion-channel studies for model development, we used pig LV for practical considerations. The qualitative agreement between the model and the RV $Ca^{2+}$ results support the robustness of the model. Further work would be of interest to characterize AP and $Ca^{2+}$ differences between RV and LV, as well as different regions in LV, in order to optimize the model for region-specific differences and determine the mechanisms.

## Conclusions

We have developed a computational model of the pig ventricular cardiomyocyte AP based on experimental data and have compared it to a human AP model. The differences between the human and porcine electrophysiological cell models suggest that conclusions from studies on arrhythmogenic mechanisms, or the prevention thereof in pigs, cannot be simply translated to humans. Through modelling, modification of biophysical processes in one species can be implemented in the other to gauge how well an arrhythmic outcome or response translates across species.

## Methods

### Ethics statement

All animal handling procedures were approved by the animal research ethics committee of the Montreal Heart Institute and conformed to principles established by the Canadian Council on Animal Care.

## Animal handling

Yorkshire X-Landrace-type pigs (2–3 months old) weighing 20 to 30 kg were anesthetized with with ketamine (20 mg/kg i.v.)/ Xylazine (2 mg/kg, i.v.) and subsequently euthanized by isoflurane (2%). The hearts were removed and transported in chilled cardioplegic solution containing (in mM): 50 $KH_2PO_4$, 8 $MgSO_4$, 10 HEPES, 5 adenosine, 140 D-glucose, 100 mannitol, pH adjusted to 7.4 with KOH. The right atrium and ventricle were rapidly removed, and the left atrium, left ventricular free wall and apex kept for cell isolation. The left anterior descending coronary artery (LAD) was cannulated through the LAD ostium, and left atrial and ventricular tissue perfused with Tyrode's solution (37˚C, 100% $O_2$). Any leaking coronary artery branches were tied off to ensure good perfusion of the tissue. Perfusion was then performed with $Ca^{2+}$-free Tyrode's solution (~10 minutes), followed by ~70-minute perfusion with the same solution containing collagenase (~0.6 mg/mL, CLSII, Worthington, Lakewood, NJ) and 0.1% bovine serum albumin (BSA, Sigma–Aldrich, Oakville, ON). Left ventricular tissues were minced and cardiomyocytes harvested. Isolated cardiomyocytes were stored in Kraftbruhe storage solution containing (mmol/L): KCl 20, $KH_2PO_4$ 10, dextrose 10, mannitol 40, L glutamic acid 70, β-OH-butyric acid10, taurine 20, and EGTA 10; 0.1% BSA, pH 7.3 with KOH, for ion-current recording.

## Patch clamp recording and $Ca^{2+}$ imaging

All in-vitro recordings were obtained at 37±2˚C. The whole-cell tight-seal patch-clamp technique was used to record currents in voltage-clamp mode. Borosilicate glass electrodes (Sutter Instruments) filled with pipette solution were connected to a patch-clamp amplifier (Axopatch 200B, Axon). Electrodes had tip resistances of 2–4 MΩ. Currents are expressed as densities (pA/pF).

Tyrode's solution contained (in mmol/L): NaCl 136, $CaCl_2$ 1.8, KCl 5.4, $MgCl_2$ 1, $NaH_2PO_4$ 0.33, dextrose 10, and HEPES 5, titrated to pH 7.4 with NaOH. For delayed-rectifier current recording, nifedipine (10-μmol/L) and 4-aminopyridine (2-mmol/L) were added to suppress $I_{CaL}$ and transient-outward current ($I_{to}$), respectively. E-4031 (5-μmol/L) was added for slow delayed rectifier ($I_{Ks}$) recording. HMR1566 (500-nmol/L) was added for rapid delayed-rectifier ($I_{Kr}$) recording. For inward-rectifier ($I_{K1}$) recording, nifedipine was replaced by $CdCl_2$ (200-μmol/L). $I_{K1}$ was recorded as the 1-mmol/L $Ba^{2+}$-sensitive current. The internal solution for $K^+$-current recording contained (in mmol/L) potassium-aspartate 110, KCl 20, $MgCl_2$ 1, MgATP 5, LiGTP 0.1, HEPES 10, sodium-phosphocreatine 5, and EGTA 5.0, titrated to pH 7.3 with KOH. The extracellular solution for $I_{Ca}$ measurement contained (in mmol/L) Tetraethylammonium chloride 136, CsCl 5.4, $MgCl_2$ 1, $CaCl_2$ 2, $NaH_2PO_4$ 0.33, dextrose 10, and HEPES 5, titrated to pH 7.4 with CsOH. Niflumic acid (50-μmol/L) was added to inhibit $Ca^{2+}$-dependent Cl-current, and 4-aminopyridine (2-mmol/L) was added to suppress $I_{to}$. The pipette solution for $I_{Ca}$-recording contained (mmol/L) CsCl 120, tetraethylammonium chloride 20, MgCl2 1, EGTA 10, MgATP 5, HEPES 10, and Li-GTP 0.1, titrated to pH 7.4 with CsOH.

## $Ca^{2+}$ imaging

The RV was dissected out of Large White castrated male pigs (35–45 kg) and perfused with an isolation solution containing (in mM) 130 NaCl, 5.4 KCl, 0.4 $NaH_2PO_4$, 1.4 $MgCl_2·6H_2O$, 5 HEPES, 10 glucose, 20 taurine, 10 creatine (pH adjusted to 7.4 with NaOH), and supplemented with 0.1 mM EGTA. Small pieces of myocardium were then agitated at 37˚C in an enzyme-containing isolation solution with collagenases (type II, Worthington, USA) and proteases (type XIV, Sigma-Aldrich). Myocytes were collected by filtration, stored at 20–23˚C, and used within 10 h.

Cells were placed in a chamber on the stage of an inverted microscope (Eclipse Ti-U, Nikon) and continuously superfused at $\sim$1 ml/min with a HEPES-based Tyrode's solution at 37˚C containing (in mmol/l) 137 NaCl, 5.4 KCl, 0.33 $NaH_2PO_4$, 0.5 $MgCl_2$·6$H_2O$, 5 HEPES, 5.6 glucose, and 1.8 $CaCl_2$ (pH adjusted to 7.4 with NaOH). Cells were field stimulated via platinum bath electrodes using a 5-ms pulse at frequencies between 1 and 5.5 Hz. $[Ca^{2+}]_i$ was monitored in isolated myocytes loaded with 3 µM fura-2 AM (Molecular probes). Myocytes were alternately illuminated with excitation light at 340 and 380 nm using a monochromator spectrophotometer system (Cairn Research). Emitted light at 510 nm was collected by a photomultiplier, and the ratio of emitted light in response to 340- and 380-nm illumination (340-to-380-nm ratio) used as an index of $[Ca^{2+}]_i$.

## Development of pig ventricular AP model

The detailed mathematical formulations and parameters of the pig ventricular AP model are provided in the S1 Text along with their provenance. The most influential parameters were fit by hand to simultaneously fit transmembrane voltage and $Ca^{2+}$ transients. The pig ventricular AP model consists of the following sarcolemmal ionic currents: fast $Na^+$ current ($I_{Na}$), Late $Na^+$ current ($I_{NaL}$), L-type $Ca^{2+}$ current ($I_{CaL}$), $Na^+$ and $K^+$ current through L-type channels ($I_{CaNa}$ and $I_{CaK}$), transient outward current due to $Ca^{2+}$-activated $Cl^-$ current, ($I_{to2}$) [18], rapid component of delayed rectifier $K^+$ current ($I_{Kr}$), slow component of delayed rectifier $K^+$ current ($I_{Ks}$), inward rectifier $K^+$ current ($I_{K1}$) and $Na^+$, $Ca^{2+}$, $K^+$ background currents ($I_{Nab}$, $I_{Cab}$, $I_{Kb}$). In addition, the model includes $Na^+$-$K^+$ ATPase current ($I_{NaK}$), $Na^+$- $Ca^{2+}$ exchanger current ($I_{NaCa}$) and sarcolemmal $Ca^{2+}$ pump current ($I_{pCa}$). The model was validated with available data on pig electrophysiological currents [8]. New experimental data on $I_{CaL}$, $I_{K1}$ and $I_{Ks}$ were also used. Intracellular $Ca^{2+}$ transient ($[Ca^{2+}]_i$) experimental data at different pacing frequencies (0.5, 1, 1.5 and 2 Hz) was used to validate model $Ca^{2+}$ transients. The finished model in several formats is included in the supplement S1 Model.

The endocardial version of O'Hara-Rudy (ORd) model was used for all comparisons and plots since it has been more thoroughly validated than the epicardial version, especially for alternans [16]. The Cardiac Arrhythmia Research Package (CARP) [35] was used for all cell simulations. C++, MATLAB (Mathworks, Natick, MA, USA) and MagicPlot (Magicplot LLC, Saint Petersburg, Russia) were used for data analyses.

## Supporting information

**S1 Text. Model equations.**
(PDF)

**S1 Model. Pig model in EasyML format for openCARP (www.openCARP.org), mmt format for the Myokit Toolkit (www.myokit.org), and CellML format (www.cellml.org).**
(ZIP)

## Acknowledgments

The authors would like to thank Dr. Michael Clerx for his help in converting the EasyML file to Myokit (www.myokit.org) mmt format.

## Author Contributions

**Conceptualization:** Stanley Nattel, Edward J. Vigmond.

**Formal analysis:** Namit Gaur.

**Funding acquisition:** Olivier Bernus, Stanley Nattel, Edward J. Vigmond.

**Investigation:** Namit Gaur, Xiao-Yan Qi, David Benoist, Ruben Coronel, Edward J. Vigmond.

**Methodology:** Namit Gaur, Xiao-Yan Qi, David Benoist, Edward J. Vigmond.

**Resources:** Stanley Nattel, Edward J. Vigmond.

**Software:** Edward J. Vigmond.

**Supervision:** Olivier Bernus, Edward J. Vigmond.

**Visualization:** Namit Gaur.

**Writing – original draft:** Namit Gaur, Stanley Nattel, Edward J. Vigmond.

**Writing – review & editing:** Namit Gaur, Olivier Bernus, Ruben Coronel, Stanley Nattel, Edward J. Vigmond.

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
