## [Decision Letter · Decision Letter 0]

19 Mar 2021

Dear Dr. Vigmond,

Thank you very much for submitting your manuscript "A Computational Model of Pig Ventricular Cardiomyocyte Electrophysiology and Calcium Handling: Translation from Pig to Human Electrophysiology" for consideration at PLOS Computational Biology.

As with all papers reviewed by the journal, your manuscript was reviewed by members of the editorial board and by several independent reviewers. In light of the reviews (below this email), we would like to invite the resubmission of a significantly-revised version that takes into account the reviewers' comments.

We cannot make any decision about publication until we have seen the revised manuscript and your response to the reviewers' comments. Your revised manuscript is also likely to be sent to reviewers for further evaluation.

Sincerely,

Daniel A Beard

Deputy Editor

PLOS Computational Biology

Reviewer's Responses to Questions

**Comments to the Authors:**

Reviewer #1: The paper by Gaur et al. describes the development and implementation of a mathematical model of the pig ventricular myocyte, simulating the electrophysiology and calcium dynamics in the cell. The model is based on species-specific data, some of which were obtained as part of this study. Importantly, the authors also compare their pig myocyte model to an established model of the human ventricular myocyte, demonstrating important differences in the ionic currents and Ca dynamics that determine the action potential, its rate dependence, and susceptibility to arrhythmias. Given the importance of the pig (a large mammal) as an experimental animal in the study of cardiac arrhythmias and effects of drug, the study is timely and important. The data supplement contains the model equations, making the model available to researchers in the field.

Several issues should be addressed by the authors:

1. Figure 2C and 2D. While the model and experiment Ca traces are qualitatively similar in showing a fast rising phase followed by a slower phase, the actual morphology is very different. The rate dependence is also different, with the peak for 0.5Hz for example at 600ms in experiment and 200ms in the model. Can the authors improve the accuracy of simulating the experimental traces? If not, the differences between model and experiment should be discussed and their effects on the results and conclusions of the study should investigated and described.

2. The Results section on alternans compares the simulations with biological experiments. What is missing in this section is comparison to the human model. You should add Panel C to Figure 8, showing simulations with the human model to demonstrate its greater sensitivity to alternans production. In the Discussion section on alternans, first paragraph, you mention APD alternans in the human model at CL=250ms and refer to Fig. 8, but Fig. 8 does not include these data.

Also, the last paragraph (Biological Experiments) is limited to Ca and does not mention the AP.

3. The SR threshold for spontaneous Ca release was lower in the pig than in the human. Thus, pig does not require as much Ca accumulation in the SR (overload) for spontaneous release as human. Contrary to your statements, this alone will make the pig more susceptible to DADs and triggered activity. The fact that the pig is LESS vulnerable to these arrhythmic events is because IK1 is much larger in the pig than in the human, providing membrane stability and also the SR Ca load at fast rate is lower in the pig.

4. Results, first paragraph, the sentence beginning with “E-4031 sensitive current….” Appears twice.

5. Rare dependence and restitution of APD. It says “….steady-state APD rate-dependence (restitution)…” --- it should be adaptation.

6. Discussion, Ionic Differences …….. Please change to…. Differences between simulated and measured ionic currents.

7. Discussion, Afterdepolarizations. The sentence “ On the other hand………….as shown by the requirement for IK1 to produce DADs in pig.” ……. It should be ...... by the requirement for 50% reduction in IK1 to produce DADs in pig.

8. Caption to Fig. 8. It will be useful to highlight (state) here that alternans were not produced in the pig, but were produced in human (after adding Panel C).

9. In Fig.8A, left, top – is the X axis time or CL - please indicate.

10. In Fig. 8A, only the 1000ms traces sow the fast – slow phase. Please comment; why is the slow phase not present at the shorter CLs?

Reviewer #2: Review of “A Computational Model of Pig Ventricular Cardiomyocyte Electrophysiology and Calcium Handling: Translation from Pig to Human Electrophysiology”

By Namit Guar, Xiao-Yan Qi, David Benoist, Olivier Bernus, Ruben Coronel, Stanley Nattel and Edward J Vigmond.

Summary: This study develops a detailed model of porcine ventricular cardiomyocyte electrophysiology for the express purpose of understanding how experiments performed with native or in culture pig cardiomyocytes can be translated to predict response in human ventricular cardiomyocytes. Experiments investigating the ion channel I-V relationships and gating in porcine cardiomyocytes from the literature plus some performed by the authors are used to parameterize each ion channel in the model. These parameterized currents are then assembled to represent the qualitative whole cell electrophysiology and calcium handling of a porcine ventricular cardiomyocyte. Then this model is used to explore the generation of early afterdepolarizations and delayed afterdepolarizations with respect to different ion channel blocks. Comparisons are made to similar ion channel blocks in the O’Hara-Rudy model of human ventricular cardiomyocytes. The authors provide a supplement detailing the model equations used and CellML code which can be run in either OpenCOR or JSim which is commendable.

Overall Comments: The model development by the authors is much needed as experiments are now currently being performed investigating the electrophysiological function of porcine cardiomyocytes which need to be translated to function in human cardiomyocytes in order to be useful. The authors make the important note that translation is not straightforward and the different mix of ion channel conductance and function must be considered. The authors fall short of actually suggesting a methodology for this translation although the summary of the differences between the two types of cardiomyocytes is valuable.

Specific Major Comments:

1) While developing an electrophysiology and ion handling model to characterize the porcine cardiomyocyte is a necessary first step for translating experimental results from pig to human, besides the observation of differences between these two models no suggestions are made on how to handle the differences for translation. What drugs might be testable in pig cardiomyocytes that would be easy to translate between species? Is the conclusion that the pig is an inadequate model for translation of any arrhythmogenic drug response to humans? Detail in the discussion of this point is lacking.

2) It is unclear in the Results, Discussion and Methods sections what experimental results on individual ion channels are taken from Verkerk et al. (Ref 8) and which experimental results were performed by the authors. In the Results, new data is stated to include INa, ICaL, IKr, IKs and IK1 experiments. In the Discussion it appears to be only ICaL, IK1 and IKs and in the Methods ICaL, IK1, IKs and IKr seem to be isolated experimentally

3) There is little discussion on how the parameters both for the individual ion channel expressions and the porcine ventricular cardiomyocyte model were obtained. The authors should explain clearly what was optimized with the different datasets. A guess from this reader would be that gating expressions and current voltage relationships were fit by hand for each ion channel. Then the whole model was assembled with these ion channels to represent porcine AP and calcium transients with only adjustments by hand to ICaL, IK1 and Ito2 max conductances being made in the model. Were other conductances or parameters optimized at the whole cardiomyocyte model level to match AP and CaT data? If a rigorous optimization was performed what methods were used and would these parameter values be unique?

Specific Minor comments

1) Abstract, line 1: “... including for mechanisms of arrhythmia.” Suggest “... including for the study of the mechanisms of arrhythmia.”

2) Abstract, general: The authors point out that translation to human myocardium is hampered by the lack of a porcine cardiomyocyte model but the conclusion is to “urge caution” when translating both computational and experimental results across species. Promising less about translation or providing some direction besides urging caution would be two different ways to handle this incongruity between the goals of the paper and the conclusion.

3) Abstract, lines 15-16: Are porcine cardiomyocytes or myocardium resistant to IKr block and sensitive to IK1 block arrhythmogenesis in experiments? If so this should be mentioned in the abstract and the Discussion.

4) Introduction, paragraph 2, line 15: “... results may not be translatable between species.”. A stronger statement would be that without computational models to understand the underlying differences between the cardiomyocytes of these two species translating results is not possible.

5) Results, paragraph 1: More detail about what parameters are adjusted to fit these current-voltage and gating datasets. Maybe a table of the parameters optimized for each ion channel would be helpful. In that table you could also identify where the data is coming from addressing some of major point 2) above. Also in figure 1 each model curve should be plotted as a continuous line instead of the piecewise linear way it is currently depicted.

6) Results, paragraph 2, line 1: How were these conductances adjusted? Were multiple datasets or just features considered during parameter adjustment Again a table would help here showing everything that was adjusted and what features or time courses were used for the optimization/hand tuning.

7) Results, paragraph 2, lines 4-9: Figure 2 referred in this section should have the same time scale on the x axis and Figure 2D should have absolute units of Ca2+ on the y axis with relative units on a second y axis. The experimental data only shows a change in florescence by ~3% so does this translate to a different percentage in absolute Ca2+? It is mentioned in the Methods that this florescence data is from RV samples from pigs. Why RV when everything else was LV?

8) Results – Rate-dependence and Restitution of APD section, paragraph 3, line 1: “The pig has lower Ca_NSR, Ca_JSR and Ca_i compared to human.”. This should be stated as a prediction made by the simulation since no experimental data is available at this resolution.

9) Results – Rate-dependence and Restitution of APD section, paragraph 3, line 2: Is the 2mM threshold for humans higher or lower than in O’Hara-Rudy and what warrants a lower threshold in the porcine cardiomyocyte? Discuss what this means here or in the Discussion section.

10) Results – Rate-dependence and Restitution of APD section, paragraph 3, line 3-8 and then paragraph 4: Are these really DADs? They are not reaching threshold in Figure 7A-D. However when the INa activation curve is shifted by leftward by 10mV DADs do occur (Figure 7E-H). There is no discussion whether this Na shift is physiologically possible. Can a drug do this? Is this just an artifact of the model that DADs can be triggered this way? In figure 7 panels A-D are not necessary.

11) Results – AP and Ca2+ transient alternans section, paragraph 1: This paragraph should be split in two with the first paragraph presenting the experimental results and the second paragraph presenting the simulation results. The simulations are stated to be qualitatively similar to the experiments but there are two noticeable differences: 1) In the simulations at 400 ms cycle length the systolic Ca2+ concentration seems to drop from the 500 ms cycle length and 2) In the simulation the Ca2+ levels in systole and diastole raise dramatically from 300 to 200 ms cycle length. It is made clear that there was no tuning of the model to fit these calcium transients however there should be some acknowledgment of these differences and discussion of what might be happening in the model and potentially physiologically. Also Figure 8 should have the same time scale on the x axis for panels in A and B. In addition an arrow to show where each higher resolution CaT is pulled from the lower resolution figure would make this figure clearer.

12) Discussion, paragraph 1, lines 5-7: “The model can be used to test potential drug therapies ... ... and allows translation of pig experimental data to human arrhythmogenesis.”. This model is the first step to understanding the differences between porcine and human cardiomyocytes but there is no discussion of how knowledge of these differences and the computational model can be used to translate experimental results from porcine myocardium to human myocardium.

13) Conclusion, paragraph 1, lines 2-6: As to minor point 11) the statement is made that experimental results from porcine myocardium cannot be simply translated to predict function in human myocardium. Can the authors say anything more about an experimental and modeling strategy for translation?

**Have all data underlying the figures and results presented in the manuscript been provided?**

Reviewer #1: Yes

Reviewer #2: Yes

PLOS authors have the option to publish the peer review history of their article (what does this mean?). If published, this will include your full peer review and any attached files.

Reviewer #1: No

Reviewer #2: **Yes: **Brian E. Carlson
---

## [Decision Letter · Decision Letter 1]

1 Jun 2021

Dear Dr. Vigmond,

We are pleased to inform you that your manuscript 'A Computational Model of Pig Ventricular Cardiomyocyte Electrophysiology and Calcium Handling: Translation from Pig to Human Electrophysiology' has been provisionally accepted for publication in PLOS Computational Biology.

Best regards,

Daniel A Beard

Deputy Editor

PLOS Computational Biology

Reviewer's Responses to Questions

**Comments to the Authors:**

Reviewer #1: The authors responded to my comments in the earlier review. One minor issue remains: the added sentence "While APD restitution was closely reproduced by the model, the changes in Ca transient shape did not occur at the same frequency, and the exact [Ca2+]i was not known." is unclear. What changes in Ca transient? The same frequency as what? Not known - in experiment or simulations? How can it be "not known" - please clarify and rewrite this sentence.

Reviewer #2: Resubmission review of “A Computational Model of Pig Ventricular Cardiomyocyte Electrophysiology and Calcium Handling: Translation from Pig to Human Electrophysiology”

By Namit Guar, Xiao-Yan Qi, David Benoist, Olivier Bernus, Ruben Coronel, Stanley Nattel and Edward J Vigmond.

Summary: This study develops a detailed model of porcine ventricular cardiomyocyte electrophysiology for the express purpose of understanding how experiments performed with native or in culture pig cardiomyocytes can be translated to predict response in human ventricular cardiomyocytes. Experiments investigating the ion channel I-V relationships and gating in porcine cardiomyocytes from the literature plus some performed by the authors are used to parameterize each ion channel in the model. These parameterized currents are then assembled to represent the qualitative whole cell electrophysiology and calcium handling of a porcine ventricular cardiomyocyte. Then this model is used to explore the generation of early afterdepolarizations and delayed afterdepolarizations with respect to different ion channel blocks. Comparisons are made to similar ion channel blocks in the O’Hara-Rudy model of human ventricular cardiomyocytes. The authors provide a supplement detailing the model equations used and CellML code which can be run in either OpenCOR or JSim which is commendable.

Overall Comments: This reviewer thanks the authors for their detailed response to comments. Thank you for clearly defining what a DAD is considered to be in your study and clarifying that translation of the observations from the computational analysis in porcine cardiomyocytes cannot be made directly to models of human cardiomyocytes without consideration of the relative differences in ion channel, transporter and receptor density and function between species.

This reviewer respectfully disagrees with the authors on one point. There are optimization strategies that: 1) are designed to fit multiple objectives and 2) can uniquely define a selected subset of model parameters. These strategies require: designing an objective function to simultaneously fit each experimental dataset (Ca2+ transient and AP at multiple pacing rates in this case), evaluating the sensitivity of the parameters in the model to this objective function, and then selecting a parameter subset for optimization of the most sensitive parameters that are not pairwise correlated to any other parameter in the model. This is a separate from the discussion of the “fidelity” of the model since all models will be an approximation to reality that are missing many components. This does not mean that an approximate model cannot fit the experimental data with a unique set of parameters.

This reviewer would omit or modify the discussion on page 12 starting with “Improving the fitting ...” and ending with “ ... 500 ms pacing?”. A more measured statement that indicates more sophisticated methods of optimization are possible but are not employed here since these preliminary hand fits to the experimental data are an important first step that suggests what may be important in translating the analysis of experimental results from pig to human.

All other comments were addressed by the authors.

**Have the authors made all data and (if applicable) computational code underlying the findings in their manuscript fully available?**

Reviewer #1: Yes

Reviewer #2: Yes

PLOS authors have the option to publish the peer review history of their article (what does this mean?). If published, this will include your full peer review and any attached files.

Reviewer #1: **Yes: **Yoram Rudy

Reviewer #2: **Yes: **Brian E. Carlson

---

## [Editor Report · Acceptance letter]

25 Jun 2021

PCOMPBIOL-D-21-00241R1 

A Computational Model of Pig Ventricular Cardiomyocyte Electrophysiology and Calcium Handling: Translation from Pig to Human Electrophysiology

Dear Dr Vigmond,

I am pleased to inform you that your manuscript has been formally accepted for publication in PLOS Computational Biology. Your manuscript is now with our production department and you will be notified of the publication date in due course.

With kind regards,

Andrea Szabo
